# FORWARD PINN: UNIFIED COMPUTATION OF SOLUTIONS AND DERIVATIVES IN A SINGLE FORWARD PASS

## ABSTRACT

Physics-Informed Neural Networks (PINNs) solve partial differential equations by embedding physical laws into their training process. A computational bottleneck, however, limits conventional PINNs. They rely on multiple backward passes to compute derivatives sequentially, a process that is memory-intensive and fails to leverage the parallelism of modern GPUs. To address this, we introduce Forward PINN, a framework that breaks from this computational architecture. Instead of relying on the backward pass, our innovation is to redesign the forward pass itself to perform differentiation. By exploiting the mathematical properties of specific activation functions, we unify the computation. The network's output and all its necessary partial derivatives—first, second, and higher-order—are calculated concurrently within a single forward propagation. This approach effectively eliminates the need for multiple backward passes for derivative computation. We validated this new architecture on the two benchmark PDE problems: the two-dimensional heat equation and the anisotropic wave equation. The experimental results show that Forward PINN achieves accuracy comparable to its conventional counterparts while delivering performance speedups of 1.55× and 1.8× on the respective benchmark problems.

## 1 INTRODUCTION

Physics-Informed Neural Networks (PINNs) (Raissi et al., 2019) have emerged as a powerful paradigm for solving partial differential equations (PDEs) by embedding physical laws directly into neural network training objectives. The fundamental principle of PINNs relies on automatic differentiation to compute partial derivatives of the network output with respect to its inputs, enabling the enforcement of governing equations as soft constraints during training (Karniadakis et al., 2021).

However, the conventional implementation of PINNs faces significant computational challenges that limit their scalability and efficiency. The standard approach requires multiple sequential operations: first, a forward pass computes the network output $u_\theta(\mathbf{x})$; subsequently, automatic differentiation is invoked to compute first-order derivatives $\frac{\partial u}{\partial x_i}$; finally, additional backward passes compute second-order derivatives $\frac{\partial^2 u}{\partial x_i \partial x_j}$ required by most PDEs (Lu et al., 2021). This sequential computation pattern presents critical limitations: **memory inefficiency** from maintaining separate computational graphs (Baydin et al., 2017), **sequential execution bottlenecks** that underutilize GPU cores and can exacerbate gradient flow issues during training (Wang et al., 2022), and **redundant graph traversals** leading to inefficient memory access patterns.

Recent efforts have explored various optimization strategies. Wang et al. (2021) proposed adaptive sampling techniques, while Krishnapriyan et al. (2021) investigated curriculum learning approaches. Self-adaptive approaches have addressed loss function balancing challenges (McClenny et al., 2023), demonstrating improved training stability through dynamic weight adjustment mechanisms. However, existing optimization methods do not address the fundamental computational inefficiency of derivative computation. More directly relevant, Bischof et al. (2008) explored forward-mode automatic differentiation for scientific computing, demonstrating theoretical advantages when the number of inputs exceeds outputs. However, these works exhibit significant limitations for PINN applications: they primarily focus on first-order derivatives while insufficiently addressing second-order

derivatives and higher-order derivatives essential for most PDE formulations, and lack specific applications to PINNs.

A fundamental gap remains: no prior work has successfully eliminated the sequential derivative computation bottleneck while maintaining the mathematical rigor and generality of the original PINN formulation.

In this paper, we introduce **Forward PINN**, a novel approach that reformulates the structure of forward propagation in PINNs. Our key insight is that by leveraging the mathematical properties of specific activation functions, we can compute network outputs and all required derivatives (first, second or higher derivatives) simultaneously in a single forward pass. This eliminates the need for multiple backward passes and their associated computational graphs, resulting in substantial improvements in both memory efficiency and computational speed.

Our main contributions are: (1) A mathematical framework that enables simultaneous calculation of solutions and derivatives in a single forward propagation process; (2) Experiments on two benchmark PDE problems demonstrating that Forward PINN achieves comparable accuracy to conventional PINNs while significantly reducing training time.

## 2 METHOD

Our Forward PINN architecture employs the Exponential Linear Unit (ELU) (Clevert et al., 2015) function as the activation function and achieves computational acceleration by rewriting the forward function to compute both solutions and partial derivatives in a single forward pass. The specific mathematical principles are as follows:

### 2.1 PROPERTY OF ELU FUNCTION

The Exponential Linear Unit (ELU) (Clevert et al., 2015) function has a property. The first-order and second-order derivatives of this function can be directly represented by its own results, without the need to input $x$. Denote the activation state as $act$. The specific representation is as follows:

$$\text{ELU}'(x) = \begin{cases} 1 & \text{if } x \geq 0 \\ \alpha e^x & \text{if } x < 0 \end{cases} = act + (1 - act) \times (\text{ELU}(x) + \alpha)$$

$$\text{ELU}''(x) = \begin{cases} 0 & \text{if } x \geq 0 \\ \alpha e^x & \text{if } x < 0 \end{cases} = (1 - act) \times (\text{ELU}(x) + \alpha)$$

$$act = \begin{cases} 1 & \text{if } x \geq 0 \\ 0 & \text{if } x < 0 \end{cases}$$

Thus, the results of the first-order and second-order derivatives at each neuron can be directly represented through the results of the forward propagation. This property is used in our algorithm.

### 2.2 CALCULATE DERIVATIVES USING FORWARD PROPAGATION

The simple feedforward neural network shown in Figure 1 is used to illustrate our algorithm. The feedforward neural network is a multilayer perceptron with two hidden layers. $h_{11}, h_{12}, ..., h_{23}$ are the results of the activation function at each neuron. The activation function is set as ELU, which is mentioned in section 2.1. Here, $[h_{11}, h_{12}, h_{13}]^T$ is denoted as $\boldsymbol{h}_1$, $[h_{21}, h_{22}, h_{23}]^T$ is denoted as $\boldsymbol{h}_2$, $[x_1, x_2]^T$ is denoted as $\boldsymbol{x}$ and $[y_1, y_2]^T$ is denoted as $\boldsymbol{y}$. The parameters for the connections between layers are defined as $\boldsymbol{W}_1, \boldsymbol{b}_1$ for the first hidden layer, $\boldsymbol{W}_2, \boldsymbol{b}_2$ for the second hidden layer, $\boldsymbol{W}_3, \boldsymbol{b}_3$ for the output layer. The process of forward propagation can be represented as follows:

$$\boldsymbol{h}_1 = \text{ELU}(\boldsymbol{W}_1\boldsymbol{x} + \boldsymbol{b}_1) \quad \boldsymbol{h}_2 = \text{ELU}(\boldsymbol{W}_2\boldsymbol{h}_1 + \boldsymbol{b}_2) \quad \boldsymbol{y} = \boldsymbol{W}_3\boldsymbol{h}_2 + \boldsymbol{b}_3$$

#### 2.2.1 FIRST-ORDER DERIVATIVE

Take $\frac{\partial \boldsymbol{y}}{\partial x_1}$ as an example. The first-order derivative of the ELU function at each neuron needs to be obtained. Denote the first-order derivative at each neuron as $h'_{11} \sim h'_{23}$. Additionally, denote

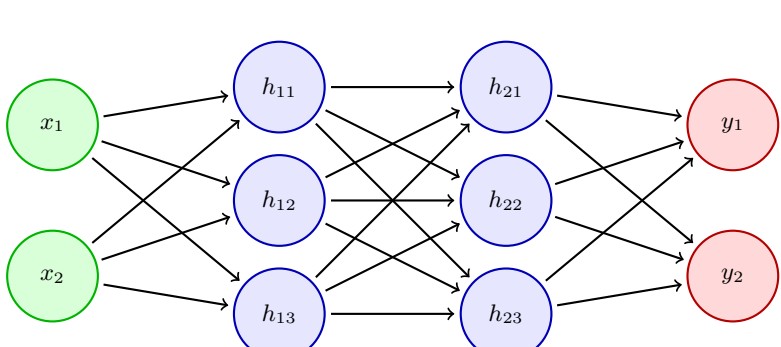

Figure 1: A multilayer perceptron architecture used to illustrate how to compute all the derivatives in a single forward pass. The network consists of an input layer with 2 neurons, two hidden layers with 3 neurons each, and an output layer with 2 neurons.

$[h'_{11}, h'_{12}, h'_{13}]^T$ as $\boldsymbol{h}'_1$ and $[h'_{21}, h'_{22}, h'_{23}]^T$ as $\boldsymbol{h}'_2$, the following can be derived:

$$\boldsymbol{h}'_1 = \text{ELU}'(\boldsymbol{W}_1\boldsymbol{x} + \boldsymbol{b}_1) \quad \boldsymbol{h}'_2 = \text{ELU}'(\boldsymbol{W}_2\boldsymbol{h}_1 + \boldsymbol{b}_2)$$

$$\frac{\partial \boldsymbol{y}}{\partial x_1} = \boldsymbol{W}_3 \times \frac{\partial \boldsymbol{h}_2}{\partial x_1} = \boldsymbol{W}_3 \times (\boldsymbol{h}'_2 \odot (\boldsymbol{W}_2 \times \frac{\partial \boldsymbol{h}_1}{\partial x_1})) = \boldsymbol{W}_3 \times (\boldsymbol{h}'_2 \odot (\boldsymbol{W}_2 \times (\boldsymbol{h}'_1 \odot (\boldsymbol{W}_1 \times \begin{pmatrix} 1 \\ 0 \end{pmatrix}))))$$

**Note:** $\odot$ is the Hadamard Product (Hadamard, 1893), specifically:

$$\begin{pmatrix} x_1 \\ x_2 \\ x_3 \end{pmatrix} \odot \begin{pmatrix} y_1 \\ y_2 \\ y_3 \end{pmatrix} = \begin{pmatrix} x_1 y_1 \\ x_2 y_2 \\ x_3 y_3 \end{pmatrix}$$

As is shown in section 2.1, $\boldsymbol{h}_1$ and $\boldsymbol{h}_2$ can be used to represent $\boldsymbol{h}'_1$ and $\boldsymbol{h}'_2$. Specifically, denote the activation state at each neuron as $act_{11} \sim act_{23}$, denote $[act_{11}, act_{12}, act_{13}]^T$ as $\boldsymbol{act}_1$ and $[act_{21}, act_{22}, act_{23}]^T$ as $\boldsymbol{act}_2$, then $\boldsymbol{h}'_1$ and $\boldsymbol{h}'_2$ can be represented as follows:

$$\boldsymbol{h}'_1 = \boldsymbol{act}_1 + (\begin{pmatrix} 1 \\ 1 \\ 1 \end{pmatrix} - \boldsymbol{act}_1) \odot (\boldsymbol{h}_1 + \begin{pmatrix} \alpha \\ \alpha \\ \alpha \end{pmatrix}) \quad \boldsymbol{h}'_2 = \boldsymbol{act}_2 + (\begin{pmatrix} 1 \\ 1 \\ 1 \end{pmatrix} - \boldsymbol{act}_2) \odot (\boldsymbol{h}_2 + \begin{pmatrix} \alpha \\ \alpha \\ \alpha \end{pmatrix})$$

In this way, the process of computing the first-order derivative of the output with respect to the input can be regarded as a kind of forward propagation process with the input being the standard basis vectors. Specifically, for a multilayer perceptron with n hidden layers, if the result of the $i$-th hidden layer is denoted as $\boldsymbol{h}_i$, the transfer function from the $i$-th layer to the $(i+1)$-th layer to compute the first-order derivative of the output to the input $x_1$ can be defined as:

$$\frac{\partial \boldsymbol{h}_{i+1}}{\partial x_1} = \boldsymbol{h}'_{i+1} \odot (\boldsymbol{W}_{i+1} \times \frac{\partial \boldsymbol{h}_i}{\partial x_1})$$

Denote $\frac{\partial \boldsymbol{h}_i}{\partial x_1}$ as $\boldsymbol{s}_i$, then the transfer function can be represented as:

$$\boldsymbol{s}_{i+1} = \boldsymbol{h}'_{i+1} \odot (\boldsymbol{W}_{i+1} \times \boldsymbol{s}_i)$$

### 2.2.2 SECOND-ORDER DERIVATIVE

Chain rule is used to construct a recursive formula, thereby we can build a model for calculating the second-order derivatives in a forward propagation way. Here, take $\frac{\partial^2 \boldsymbol{y}}{\partial x_1^2}$ as an example. With chain rule, the following can be derived:

$$\frac{\partial^2 \boldsymbol{y}}{\partial x_1^2} = \boldsymbol{W}_3 \times \frac{\partial^2 \boldsymbol{h}_2}{\partial x_1^2}$$

To calculate $\frac{\partial^2 \boldsymbol{h}_2}{\partial x_1^2}$, the second-order derivative of the ELU function at each neuron needs to be obtained. Similar to the representation used when computing the first-order derivative, the second-order derivative at each neuron is denoted as $h_{11}'' \sim h_{23}''$. Additionally $[h_{11}'', h_{12}'', h_{13}'']^T$ is denoted as $\boldsymbol{h}_1''$, $[h_{21}'', h_{22}'', h_{23}'']^T$ is denoted as $\boldsymbol{h}_2''$, then the following can be derived:

$$\boldsymbol{h}_1'' = \mathrm{ELU}''(\boldsymbol{W}_1 \boldsymbol{x} + \boldsymbol{b}_1) \quad \boldsymbol{h}_2'' = \mathrm{ELU}''(\boldsymbol{W}_2 \boldsymbol{h}_1 + b_2)$$

Shown in section 2.1, $\boldsymbol{h}_1''$ and $\boldsymbol{h}_2''$ can be represented with $\boldsymbol{h}_1$ and $\boldsymbol{h}_2$, To be specific:

$$\boldsymbol{h}_1'' = (\begin{pmatrix} 1 \\ 1 \\ 1 \end{pmatrix} - \boldsymbol{act}_1) \odot (\boldsymbol{h}_1 + \begin{pmatrix} \alpha \\ \alpha \\ \alpha \end{pmatrix}) \quad \boldsymbol{h}_2'' = (\begin{pmatrix} 1 \\ 1 \\ 1 \end{pmatrix} - \boldsymbol{act}_2) \odot (\boldsymbol{h}_2 + \begin{pmatrix} \alpha \\ \alpha \\ \alpha \end{pmatrix})$$

As is shown in section 2.2.1:

$$\frac{\partial \boldsymbol{h}_2}{\partial x_1} = \boldsymbol{h}_2' \odot (\boldsymbol{W}_2 \times \frac{\partial \boldsymbol{h}_1}{\partial x_1}) = \mathrm{ELU}'(\boldsymbol{W}_2 \boldsymbol{h}_1 + \boldsymbol{b}_2) \odot (\boldsymbol{W}_2 \times \frac{\partial \boldsymbol{h}_1}{\partial x_1})$$

According to the Leibniz formula, the following can be derived:

$$\frac{\partial^2 \boldsymbol{h}_2}{\partial x_1^2} = \mathrm{ELU}''(\boldsymbol{W}_2 \boldsymbol{h}_1 + \boldsymbol{b}_2) \odot (\boldsymbol{W}_2 \times \frac{\partial \boldsymbol{h}_1}{\partial x_1}) \odot (\boldsymbol{W}_2 \times \frac{\partial \boldsymbol{h}_1}{\partial x_1}) + \mathrm{ELU}'(\boldsymbol{W}_2 \boldsymbol{h}_1 + \boldsymbol{b}_2) \odot (\boldsymbol{W}_2 \times \frac{\partial^2 \boldsymbol{h}_1}{\partial x_1^2})$$

Denote $\frac{\partial^2 \boldsymbol{h}_i}{\partial x_1^2}$ as $\boldsymbol{t}_i$, and $\frac{\partial \boldsymbol{h}_i}{\partial x_1}$ as $\boldsymbol{s}_i$, then the formula above can be denoted as follows:

$$\boldsymbol{t}_2 = \boldsymbol{h}_2'' \odot (\boldsymbol{W}_2 \times \boldsymbol{s}_1) \odot (\boldsymbol{W}_2 \times \boldsymbol{s}_1) + \boldsymbol{h}_2' \odot (\boldsymbol{W}_2 \times \boldsymbol{t}_1)$$

Generally, the transfer function from the i-th layer to the (i+1)-th layer to compute the second-order derivative of the output to the input $x_1$ can be defined as:

$$\boldsymbol{t}_{i+1} = \boldsymbol{h}_{i+1}'' \odot (\boldsymbol{W}_{i+1} \times \boldsymbol{s}_i) \odot (\boldsymbol{W}_{i+1} \times \boldsymbol{s}_i) + \boldsymbol{h}_{i+1}' \odot (\boldsymbol{W}_{i+1} \times \boldsymbol{t}_i)$$

Similarly, To calculate $\frac{\partial^2 \boldsymbol{y}}{\partial x_1 \partial x_2}$, the following can be derived:

$$\frac{\partial^2 \boldsymbol{h}_2}{\partial x_1 \partial x_2} = \mathrm{ELU}''(\boldsymbol{W}_2 \boldsymbol{h}_1 + \boldsymbol{b}_2) \odot (\boldsymbol{W}_2 \times \frac{\partial \boldsymbol{h}_1}{\partial x_2}) \odot (\boldsymbol{W}_2 \times \frac{\partial \boldsymbol{h}_1}{\partial x_1}) + \mathrm{ELU}'(\boldsymbol{W}_2 \boldsymbol{h}_1 + \boldsymbol{b}_2) \odot (\boldsymbol{W}_2 \times \frac{\partial^2 \boldsymbol{h}_1}{\partial x_1 \partial x_2})$$

Denote $\frac{\partial^2 \boldsymbol{h}_i}{\partial x_1 \partial x_2}$ as $\boldsymbol{t}_i$, $\frac{\partial \boldsymbol{h}_i}{\partial x_1}$ as $\boldsymbol{s}_{i_1}$, $\frac{\partial \boldsymbol{h}_i}{\partial x_2}$ as $\boldsymbol{s}_{i_2}$, then the transfer function from the $i$-th layer to the $(i+1)$-th layer to compute the second-order derivative of the output to the input $x_1$ and $x_2$ can bedefined as:

$$\boldsymbol{t}_{i+1} = \boldsymbol{h}_{i+1}'' \odot (\boldsymbol{W}_{i+1} \times \boldsymbol{s}_{i_2}) \odot (\boldsymbol{W}_{i+1} \times \boldsymbol{s}_{i_1}) + \boldsymbol{h}_{i+1}' \odot (\boldsymbol{W}_{i+1} \times \boldsymbol{t}_i)$$

### 2.2.3 Summary of the Forward Algorithm

Above all, the process of computing first-order derivative and second-order derivative can be regarded as a kind of forward propagation. Specifically, define the output of the $i$-th hidden layer as $\boldsymbol{h}_i$, And denote $\frac{\partial \boldsymbol{h}_i}{\partial x_j}$ as $\boldsymbol{s}_{ij}$, $\frac{\partial^2 \boldsymbol{h}_i}{\partial x_j \partial x_k}$ as $\boldsymbol{t}_{ijk}$, the forward propagation between the $i$-th layer and the $(i+1)$-th layer ($i \geq 0$) can be defined as:

$$\boldsymbol{h}_{i+1} = \mathrm{ELU}(\boldsymbol{W}_{i+1} \boldsymbol{h}_i + \boldsymbol{b}_{i+1})$$

$$\boldsymbol{h}_{i+1}' = \boldsymbol{act}_{i+1} + (\boldsymbol{1}_{i+1} - \boldsymbol{act}_{i+1}) \odot (\boldsymbol{h}_{i+1} + \alpha \times \boldsymbol{1}_{i+1})$$

$$\boldsymbol{h}_{i+1}'' = (\boldsymbol{1}_{i+1} - \boldsymbol{act}_{i+1}) \odot (\boldsymbol{h}_{i+1} + \alpha \times \boldsymbol{1}_{i+1})$$

$$\boldsymbol{s}_{(i+1)j} = \boldsymbol{h}_{i+1}' \odot (\boldsymbol{W}_{i+1} \times \boldsymbol{s}_{ij})$$

$$\boldsymbol{t}_{(i+1)jk} = \boldsymbol{h}_{i+1}'' \odot (\boldsymbol{W}_{i+1} \times \boldsymbol{s}_{ij}) \odot (\boldsymbol{W}_{i+1} \times \boldsymbol{s}_{ik}) + \boldsymbol{h}_{i+1}' \odot (\boldsymbol{W}_{i+1} \times \boldsymbol{t}_{ijk})$$

$\boldsymbol{1}_i$ is the vector of ones , of which the dimension is the same as $\boldsymbol{h}_i$. Specifically, for $i = 0$, $\boldsymbol{h}_0 = \boldsymbol{x}$, $\boldsymbol{s}_{0j} = \boldsymbol{e}_j$, $\boldsymbol{t}_{0jk} = \boldsymbol{0}$. $\boldsymbol{x}$ is the input of the MLP, $\boldsymbol{e}_j$ is the standard basis vector with 1 in the $j$-th position and 0 elsewhere. $\boldsymbol{0}$ is the zero vector (all elements are 0).

With these formulas, we can calculate the output, the first-order derivative and the second-order derivative at the same time, just using forward propagation.

**Remark 1:** The forward propagation framework presented above is not limited to the ELU activation function. Any activation function with known analytical expressions for its first-order derivative $h'_{i+1}$ and second-order derivative $h''_{i+1}$ can be incorporated into this framework, enabling efficient derivative computation.

**Remark 2:** While we have demonstrated the computation of first- and second-order derivatives, the proposed method can be extended to higher-order derivatives. By applying the chain rule recursively to the second-order derivative terms $t_{(i+1)jk}$, one can derive the update formulas for third-order and higher-order derivatives following the same forward propagation principle.

### 2.3 ADVANTAGE ANALYSIS

The conventional Physics-Informed Neural Network(PINN)(Raissi et al., 2019) employs a fully-connected neural network to approximate the solution u, and record the computation graph of forward propagation at the same time. Then the computation graph is used to calculate the first-order derivative. Similarly, the computation graph is recorded to enable second-order differentiation while computing the first-order derivatives. The overall computation graph is shown in Figure 2 (a). The computation graph illustrates a propagation process for a multilayer perceptron with three hidden layers when the conventional PINN approach is employed. It shows how the output, first-order derivatives and second-order derivatives are computed through forward and backward propagation.

The Forward PINN also employs a fully-connected neural network to approximate the solution u. However, it doesn't need computation graph to calculate first-order or second-order derivatives. Derivatives can be calculated just using forward propagation. The overall computation graph is shown in Figure 2 (b).

Comparing the two computation graphs, the advantages of Forward PINN are as follows:

1. The propagation path to calculate second-order derivative is much shorter, approximately half the length of that required by conventional algorithms.

2. Forward PINN stores only a single computational graph for network-parameter optimization, whereas the conventional algorithm must maintain at least three separate graphs: one for calculating first-order derivatives, one for calculating second-order derivatives and one for parameters updation. This approach substantially reduces memory consumption.

3. Forward PINN is ideally suited to the GPU's parallel architecture(NVIDIA Corporation, 2020): its computational density(the ratio of arithmetic operations to global memory accesses) is significantly higher. Because the output, first-order derivatives and second-order derivatives are computed simultaneously within the same forward kernel.

   In contrast, the conventional PINN compute these quantities sequentially, which forces multiple, memory-intensive passes over the network. This sequential pattern under-utilizes GPU cores and incurs repeated global-memory traffic, making it inherently less efficient on modern accelerators. As noted by Hooker (2021), the evolution of machine learning algorithms has been significantly shaped by hardware constraints, particularly the parallel architecture of GPUs, emphasizing the importance of designing algorithms that align with hardware capabilities rather than fighting against them.

## 3 EXPERIMENT DESIGN

The experiments were conducted in two scenarios: one based on the two-dimensional heat equation, and the other based on the anisotropic wave equation.

### 3.1 SCENARIO ONE

Scenario One is based on two-dimensional heat equation(Raissi et al., 2019; Lu et al., 2021), the physical constraints can be described as follows:

$$\frac{\partial u}{\partial t} = \alpha(\frac{\partial^2 u}{\partial x^2} + \frac{\partial^2 u}{\partial y^2})$$

(a) Computation graph for conventional PINN

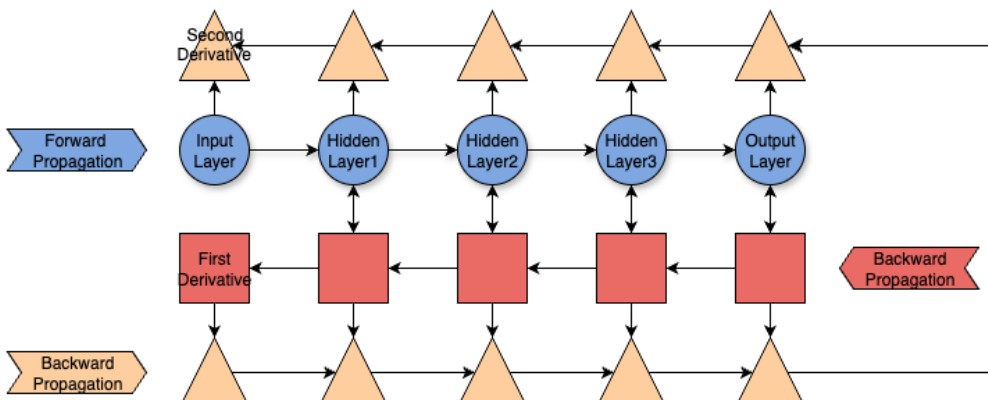

(b) Computation graph for Forward PINN

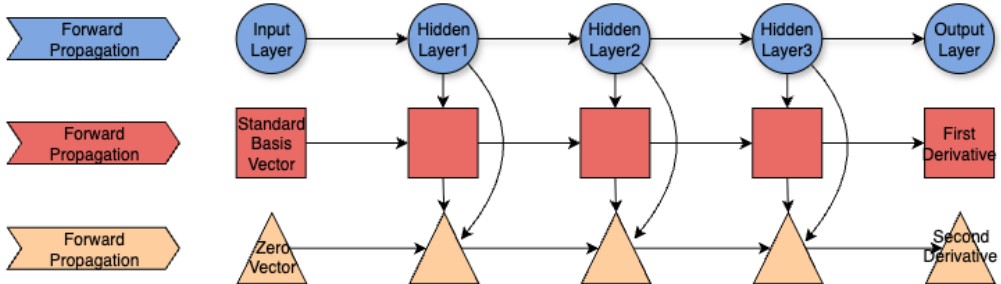

Figure 2: Computation graph to help analyze the advantages: (a) for conventional PINN and (b) for Forward PINN. Circles denote the nodes during the forward process to compute solutions. Squares represent the nodes where the first-order derivatives are computed. Triangles mark the nodes involved in computing the second-order derivatives. The computation graph ignores the details of operations, instead emphasizes the routes taken by forward or backward propagation. Arrows between nodes indicate the direction of data flow.

u(t,x,y) is the temperature distribution function, $\alpha$ is the thermal diffusivity constant, which is set as 0.01 in this experiment. The domain is defined as $(t, x, y) \in [0, 1] \times [0, 1] \times [0, 1]$.

Experimental data were generated using the analytical solution for diffusion from a Gaussian heat source(Polyanin & Zaitsev, 2002; Özişik, 1993). The expression for the solution is as follows:

$$u(t, x, y) = \frac{1}{2\pi\sigma_t^2} \exp\left(-\frac{(x - x_0)^2 + (y - y_0)^2}{2\sigma_t^2}\right)$$

$(x_0, y_0) = (0.5, 0.5)$ is the center position of the heat source, $\sigma_t = \sqrt{\sigma_0^2 + 2\alpha t}$ is the time-dependent diffusion width, $\sigma_0 = 0.1$ is the initial width of the heat source.

Specifically, 20 time points are uniformly selected in the interval (0, 1). For each time point, a 20 ×20 grid is constructed to cover the region (0, 1) × (0, 1), resulting in 8,000 spatiotemporal points. For each spatiotemporal point, the analytical solution of the Gaussian heat source diffusion is used as the ground truth, and maximum normalization is applied to construct the training set.

## 3.2 SCENARIO TWO

Scenario Two is based on anisotropic wave equation (Raissi et al., 2019; McClenny et al., 2023), the physical constraints can be described as follows:

$$\frac{\partial^2 u}{\partial t^2} = c1^2 \times \frac{\partial^2 u}{\partial x^2} + c2^2 \times \frac{\partial^2 u}{\partial y^2} + c3^2 \times \frac{\partial^2 u}{\partial x \partial y}$$

u(t, x, y) is the wave field function, $c_1 = 1.0$ is the wave speed parameter in the $x$ direction, $c_2 = 0.8$ is the wave speed parameter in the $y$ direction, and $c_3 = 0.3$ is the coupling coefficient for the cross term. The domain is defined as $(t, x, y) \in [0, 1] \times [-1, 1] \times [-1, 1]$.

Experimental data were generated using the analytical solution for anisotropic wave propagation from a Gaussian wave packet(Brillouin, 1946; Gabor, 1946). The mathematical expression for the solution is as follows:

$$u(t, x, y) = \exp\left(-\frac{1}{2}r^2\right) \times \sin(\omega t - k_1 x - k_2 y - k_3 xy)$$

where $r = \sqrt{(c_1 x)^2 + (c_2 y)^2 + c_3 xy}$ is the anisotropic distance function, $\omega = \sqrt{c_1^2 + c_2^2 + c_3^2}$ is the angular frequency, $k_1 = c_1$, $k_2 = c_2$, $k_3 = c_3$ are the wave numbers in different directions.

Similar to Scenario One, 20 time points are uniformly selected in the interval (0, 1). For each time point, a 20 $\times$20 grid is constructed to cover the region (-1, 1) $\times$ (-1, 1), resulting in 8,000 spatiotemporal points. For each spatiotemporal point, the analytical solution of the anisotropic wave propagation is used as the ground truth, and maximum normalization is applied to construct the training set.

## 3.3 COMPARISON EXPERIMENT DESIGN

To compare the performance of the conventional PINN and the proposed Forward PINN, both methods are evaluated under the two scenarios described. For each scenario, the same feedforward neural network, training data, and optimizer settings are used to ensure a fair comparison between methods.

Specifically, a feedforward neural network with an input dimension of 3, an output dimension of 1, four hidden layers each containing 64 neurons, and ELU activation functions is used to fit the input and output of the training set. Both methods use the Adam optimizer with a learning rate of 0.001.

In the PyTorch framework, both the conventional PINN and our proposed Forward PINN methods were implemented and trained on a single NVIDIA A100 GPU for 50 iterations. For the Standard PINN, automatic differentiation provided by PyTorch was used to compute the required first and second-order derivatives during training. This involves recording the computation graph during the forward pass and performing multiple backward passes to obtain the derivatives. For the Forward PINN, the forward propagation formulas described in Section 2.2 were implemented to compute the output, first-order, and second-order derivatives simultaneously in a single forward pass, without relying on PyTorch's autograd for derivatives. The pseudocode for both algorithms is in Appendix B.

# 4 RESULTS

This section presents a comprehensive evaluation of the Forward PINN framework. The empirical analysis is structured to validate three critical aspects of the proposed method: the numerical accuracy of its derivative computations, its training efficiency relative to conventional PINNs, and the quality of the final solutions it produces.

## 4.1 DERIVATIVE COMPUTATION ACCURACY VALIDATION

A preliminary validation was conducted to rigorously assess the numerical accuracy of the proposed forward-mode derivative computation. At each step of the training process, the first and second-order partial derivatives computed via the Forward PINN algorithm were quantitatively compared against those obtained from PyTorch's established automatic differentiation (`autograd`) engine.

The comparative analysis revealed excellent agreement between the two methods. For derivative values typically in the range of $10^{-2}$ to $10^{-3}$, the discrepancies between our Forward PINN method and PyTorch's `autograd` were consistently on the order of $10^{-9}$, demonstrating machine-precision accuracy. This outcome not only confirms the mathematical integrity and correct implementation of our approach but also validates that our method produces results virtually identical to the established automatic differentiation framework.

## 4.2 TRAINING EFFICIENCY COMPARISON

The training efficiency of the Forward PINN and the conventional PINN was compared in terms of convergence speed and training time. For the convergence speed, the loss curves of both methods for two scenarios in the first 50 training iterations are shown in Figure 3. The loss curves indicate that both methods converge at a similar speed, confirming the correctness of our algorithm.

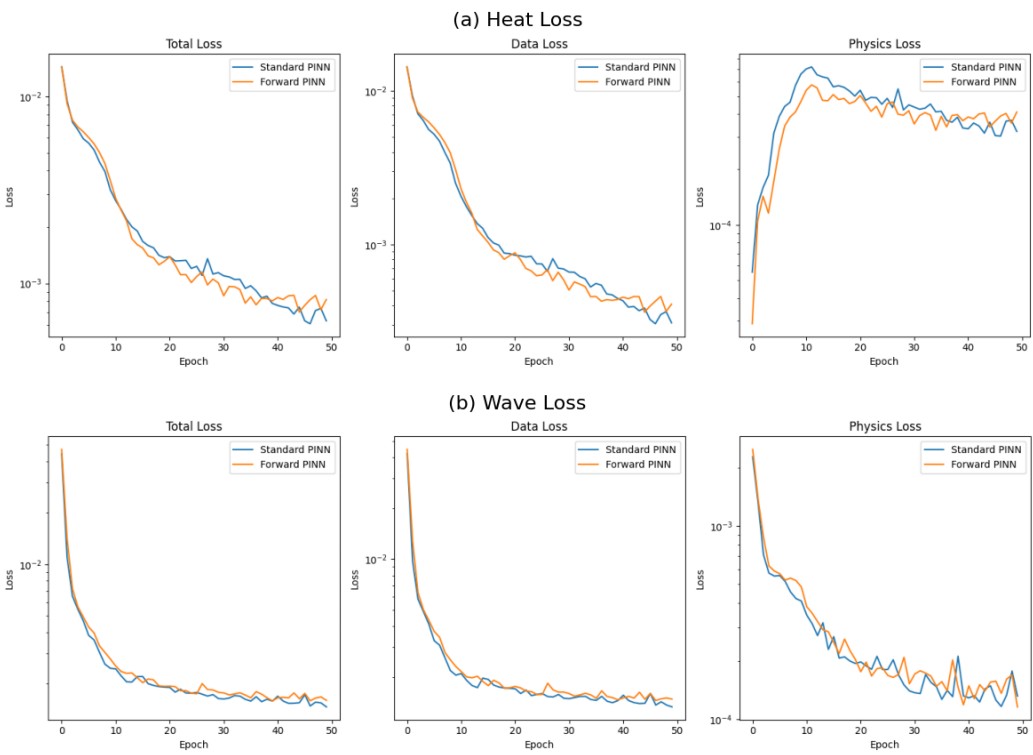

Figure 3: Loss curves comparing the two methods: (a) for Scenario One and (b) for Scenario Two. Both methods demonstrate similar convergence behavior.

For the training time, the training time for each epoch and the total training time for both methods across two scenarios are shown in Figure 4

For Scenario One, the total training time for the Forward PINN is 32.32seconds, while the conventional PINN takes 50.24 seconds, indicating a significant speedup of approximately 1.55 times. For Scenario Two, the total training time for the Forward PINN is 42.85 seconds, while the conventional PINN takes 77.21 seconds, indicating a significant speedup of approximately 1.8 times. The results demonstrate that the Forward PINN has a significant advantage in training efficiency.

## 4.3 TRAINING EFFECT COMPARISON

To ensure that the observed efficiency gains did not compromise the quality of the final solution, the predictive accuracy of both methods was evaluated against the known analytical solutions. The

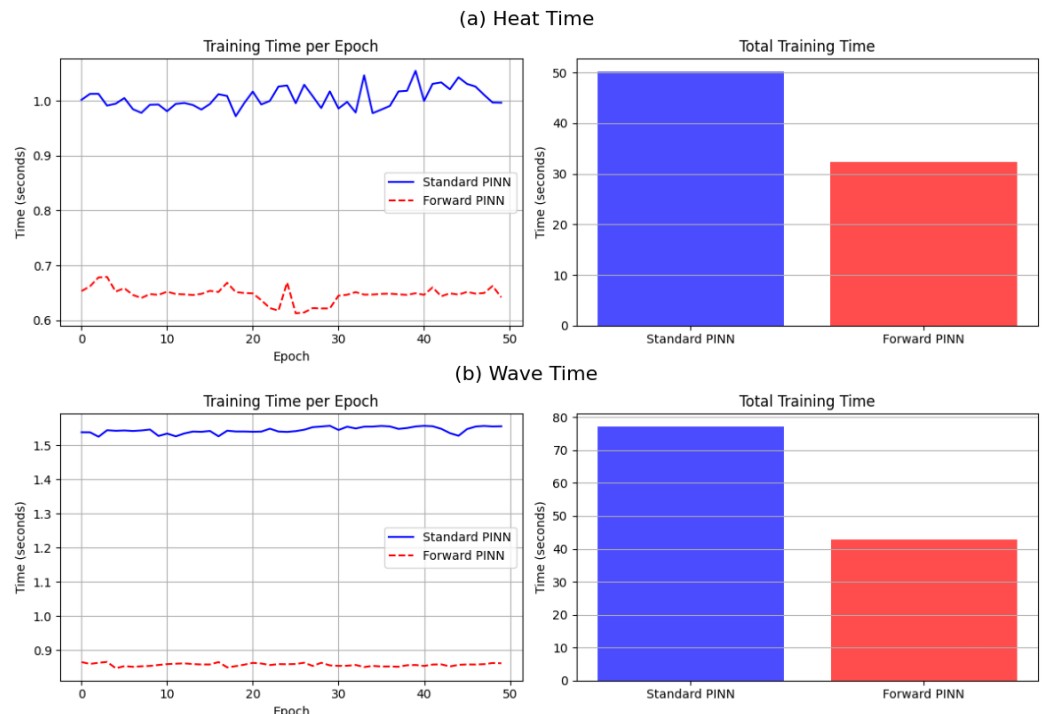

Figure 4: Training time for each epoch and total training time for both methods: (a) for Scenario One and (b) for Scenario Two. The Forward PINN is approximately 1.5 times faster than the conventional PINN in Scenario One and 1.8 times faster in Scenario Two.

temporal evolution of the prediction error was visualized through animated GIFs (available in the supplementary materials), with key time-slice comparisons provided in Appendix C.

This qualitative analysis confirms that both the Forward PINN and the conventional PINN produce solutions of a similar high fidelity when compared to the ground truth. The error distributions over the spatiotemporal domain are visually indistinguishable between the two methods, affirming that the Forward PINN architecture successfully learns the physical dynamics with an accuracy comparable to its traditional counterpart.

## 5 CONCLUSION

In this paper, we addressed a critical computational bottleneck in conventional Physics-Informed Neural Networks (PINNs) arising from their reliance on sequential backward passes for derivative computation. We introduced **Forward PINN**, a novel framework that reformulates the network's forward pass to compute the solution and its required partial derivatives simultaneously. By leveraging the analytical properties of activation functions like ELU, our method eliminates the need for multiple, memory-intensive autograd calls, creating a more efficient, unified computational pipeline.

Our experimental results on the 2D heat and anisotropic wave equations validate the effectiveness of our approach. We demonstrated that Forward PINN achieves accuracy and convergence rates comparable to conventional PINNs while delivering significant performance improvements, with training speedups of up to $1.8\times$. These findings confirm that our method successfully reduces computational overhead without compromising the accuracy or training stability of the PINN framework.

The primary implication of this work is a more scalable and hardware-efficient formulation of PINNs. By reducing memory footprint and aligning the computational pattern with the parallel architecture of modern GPUs, Forward PINN paves the way for tackling larger and more complex scientific machine learning problems.

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

## A  LLM USAGE DISCLOSURE

In accordance with ICLR 2026 guidelines, we disclose the use of (LLMs) in this research:

**Writing Assistance:** LLMs were used to assist in drafting portions of the abstract and introduction sections. The authors provided the core research ideas, technical content, and scientific contributions, while LLMs helped with language refinement, structure organization, and clarity improvement.

**Code Development:** LLMs were utilized to generate code components for:

- Implementation of the conventional PINN baseline model
- Data visualization scripts for comparative analysis between Forward PINN and conventional PINN
- Utility functions for performance benchmarking and result plotting

All generated code was thoroughly reviewed, tested and modified by the authors to ensure correctness and alignment with the research objectives. The core algorithmic contributions, mathematical derivations, and experimental design were conceived and implemented by the human authors.

## B  PSEUDOCODE FOR BOTH ALGORITHMS

**Conventional PINN:**

1. Initialize neural network.
2. For each training iteration:
   (a) Perform forward propagation to compute the network output $u_\theta(\boldsymbol{x})$.
   (b) Use automatic differentiation (torch.autograd.grad() function, with both create_graph and retain_graph parameters set to True) to compute the first-order derivatives $\frac{\partial u}{\partial t}$, $\frac{\partial u}{\partial x}$, and $\frac{\partial u}{\partial y}$. Here manual optimization is used to calculate the three derivatives simultaneously.
   (c) Use automatic differentiation again to compute the required second-order derivatives
   (d) Compute the loss function using the solutions and derivatives.
   (e) Update parameters $\theta$ using the optimizer.

**Forward PINN:**

Note:To reduce memory usage, the pseudocode occasionally reuses the same variable name on both sides of the assignment, here the equals sign denotes programming assignment, not mathematical equality.

1. Initialize neural network.
2. Redefine forward function of the network to compute output, first-order derivatives, and second-order derivatives using the formulas described in Section 2.2 at the same time:
   (a) Get the input and reshape them into the vector form $[t, x, y]^T$.
   (b) Create input for first-order derivative calculation:
   $$\boldsymbol{e}_1 = [1, 0, 0]^T, \quad \boldsymbol{e}_2 = [0, 1, 0]^T, \quad \boldsymbol{e}_3 = [0, 0, 1]^T$$
   where $\boldsymbol{e}_1$ is for $\frac{\partial u}{\partial t}$, $\boldsymbol{e}_2$ is for $\frac{\partial u}{\partial x}$, and $\boldsymbol{e}_3$ is for $\frac{\partial u}{\partial y}$.
   (c) Create input for second-order derivative calculation: $\boldsymbol{z} = [0, 0, 0]^T$
   (d) Let $\boldsymbol{u} = [t, x, y]^T$, $\boldsymbol{u}_t = \boldsymbol{e}_1$, $\boldsymbol{u}_x = \boldsymbol{e}_2$, $\boldsymbol{u}_y = \boldsymbol{e}_3$, $\boldsymbol{u}_{xx} = \boldsymbol{u}_{xy} = \boldsymbol{u}_{yy} = \boldsymbol{u}_{tt} = \boldsymbol{z}$.
   (e) For each layer $i$ in the network:
       i. Get the weight matrix $\boldsymbol{W}_i$ and bias vector $\boldsymbol{b}_i$ for the layer.
       ii. Compute the pre-activation values: $\boldsymbol{z}_i = \boldsymbol{W}_i \boldsymbol{u} + \boldsymbol{b}_i$
       iii. Apply ELU activation: $\boldsymbol{u} = \text{ELU}(\boldsymbol{z}_i)$

iv. Construct activation state vector by element-wise comparison with zero:

$$\boldsymbol{act}_i[j] = \begin{cases} 1 & \text{if } \boldsymbol{z}_i[j] \geq 0 \\ 0 & \text{if } \boldsymbol{z}_i[j] < 0 \end{cases}$$

where $j$ indexes each dimension of the layer output.

v. Compute first-order derivative coefficients:

$$\boldsymbol{u}_i' = \boldsymbol{act}_i + (\boldsymbol{1} - \boldsymbol{act}_i) \odot (\boldsymbol{u} + \alpha \cdot \boldsymbol{1})$$

vi. Compute second-order derivative coefficients:

$$\boldsymbol{u}_i'' = (\boldsymbol{1} - \boldsymbol{act}_i) \odot (\boldsymbol{u} + \alpha \cdot \boldsymbol{1})$$

vii. Multiply first-order derivatives with weight matrix:

$$\boldsymbol{u}_t = (\boldsymbol{W}_i \times \boldsymbol{u}_t), \quad \boldsymbol{u}_x = (\boldsymbol{W}_i \times \boldsymbol{u}_x), \quad \boldsymbol{u}_y = (\boldsymbol{W}_i \times \boldsymbol{u}_y)$$

viii. Update second-order derivatives:

$$\boldsymbol{u}_{tt} = \boldsymbol{u}_i'' \odot \boldsymbol{u}_t \odot \boldsymbol{u}_t + \boldsymbol{u}_i' \odot (\boldsymbol{W}_i \times \boldsymbol{u}_{tt})$$

$$\boldsymbol{u}_{xx} = \boldsymbol{u}_i'' \odot \boldsymbol{u}_x \odot \boldsymbol{u}_x + \boldsymbol{u}_i' \odot (\boldsymbol{W}_i \times \boldsymbol{u}_{xx})$$

$$\boldsymbol{u}_{yy} = \boldsymbol{u}_i'' \odot \boldsymbol{u}_y \odot \boldsymbol{u}_y + \boldsymbol{u}_i' \odot (\boldsymbol{W}_i \times \boldsymbol{u}_{yy})$$

$$\boldsymbol{u}_{xy} = \boldsymbol{u}_i'' \odot \boldsymbol{u}_x \odot \boldsymbol{u}_y + \boldsymbol{u}_i' \odot (\boldsymbol{W}_i \times \boldsymbol{u}_{xy})$$

ix. Update the first-order derivatives:

$$\boldsymbol{u}_t = \boldsymbol{u}_i' \odot \boldsymbol{u}_t$$

$$\boldsymbol{u}_x = \boldsymbol{u}_i' \odot \boldsymbol{u}_x$$

$$\boldsymbol{u}_y = \boldsymbol{u}_i' \odot \boldsymbol{u}_y$$

(f) Extract final outputs: solution $\boldsymbol{u}$, and derivatives $\boldsymbol{u}_t, \boldsymbol{u}_x, \boldsymbol{u}_y, \boldsymbol{u}_{xx}, \boldsymbol{u}_{yy}, \boldsymbol{u}_{xy}, \boldsymbol{u}_{tt}$.

3. For each training iteration:

   (a) Call the redefined forward function to get solutions and all required derivatives.
   (b) Compute the physics-informed loss function using the solutions and derivatives.
   (c) Update network parameters using backward propagation.

## C   THE SLICES OF THE GIF ANIMATION WHICH IS CREATED TO SHOW THE EVOLUTION OF THE PREDICTED SOLUTION ERROR OVER TIME

These two figures demonstrate that Forward PINN and conventional PINN achieve very similar performance on the test set, providing additional evidence for the correctness of our proposed algorithm.

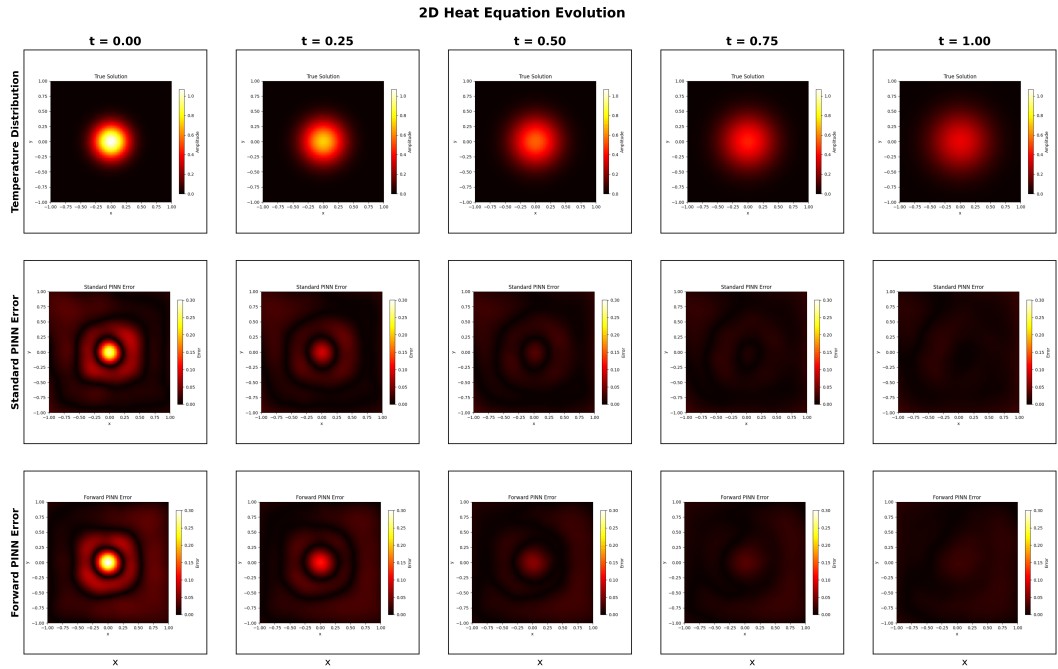

Figure 5: Slices of the GIF animation showing the predicted solution error at different time points for Scenario One. The ground truth is shown in the first row, the predicted solution error by the Standard PINN is shown in the second row, and the predicted solution error by the Forward PINN is shown in the third row. The error is shown in the fourth row. The error is calculated as the absolute difference between the predicted solution and the ground truth.

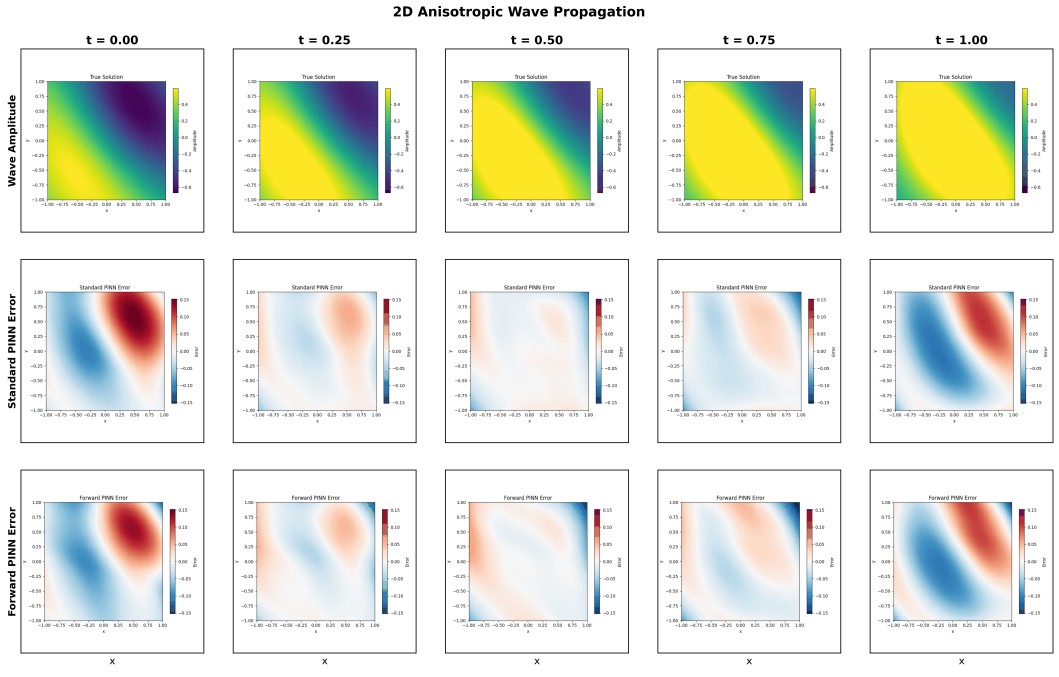

Figure 6: Slices of the GIF animation showing the predicted solution error at different time points for Scenario Two. The visualization is presented in the same way as in Figure 5.