# OpenReview forum: "Forward PINN: Unified Computation of Solutions and Derivatives in a Single Forward Pass"
_ICLR.cc/2026/Conference — ICLR 2026 Conference Desk Rejected Submission_

### Official Review · Reviewer_2FWx · 2025-10-26

**Soundness:** 1
**Presentation:** 2
**Contribution:** 1
**Rating:** 0
**Confidence:** 5

**Summary:**

The paper proposes Forward PINN, which attempts to compute both network outputs and their derivatives within a single forward pass by exploiting analytical properties of the ELU activation function. The authors claim this removes the need for multiple backward passes in conventional PINNs, improving computational efficiency. Experiments are limited to two toy PDEs (2-D heat equation and a simple anisotropic wave equation), showing modest speedups with similar accuracy compared to the baseline PINN.

**Strengths:**

The paper is clearly written and implements a straightforward idea that can be reproduced easily; it may serve as a pedagogical example for beginners exploring the computational pipeline of PINNs.

**Weaknesses:**

- **Lack of novelty and contribution:**
  The core idea—replacing automatic differentiation with explicit derivative formulas for specific activations—is a basic forward-mode differentiation trick, not a new mathematical or algorithmic framework.

- **Trivial problem setup:**
  Experiments are restricted to extremely simple PDEs (heat and wave equations with analytical solutions) that provide no evidence of generality, robustness, or scalability.

- **No theoretical foundation:**
  There is no rigorous analysis, convergence argument, or error estimate to justify that the proposed forward derivative propagation is mathematically consistent or stable for training PINNs.

- **Weak experimental validation:**
  Results only report minor speed improvements and comparable accuracy; no ablation, memory analysis, or comparison.


Overall, the work does not reach the level of a research contribution.

**Questions:**

The main questions for the authors are already reflected in the **Weaknesses** section above, as they directly arise from the identified issues.

---

### Official Review · Reviewer_j24u · 2025-10-26

**Soundness:** 1
**Presentation:** 2
**Contribution:** 3
**Rating:** 4
**Confidence:** 4

**Summary:**

The paper introduces Forward PINN, a method for efficiently computing neural network outputs and their derivatives using only a single forward pass. The approach applies to any network architecture because it utilises the analytical derivatives of activation functions and the manner in which these derivatives propagate through the network. A key advantage of the method is that it eliminates the need for multiple backward passes to compute higher-order derivatives with respect to the input, resulting in improved memory efficiency and faster training.
Finally, the authors apply the method for Physics-Informed Neural Networks training to two benchmark problems: the heat and wave equations. Their results show that the analytical evaluation produces results identical to automatic differentiation (up to machine precision), while achieving a speedup compared to the standard automatic differentiation method.

**Strengths:**

The paper addresses an important problem in the training of physics-informed neural networks, focusing on computational bottlenecks such as memory inefficiency, sequential execution that limits GPU parallelism, and redundant graph traversal when computing higher-order derivatives.

The proposed algorithmic contribution is original and broadly applicable, provided that the activation derivatives are analytical, and it is clearly explained. The experimental section demonstrates that the computation performed by automatic differentiation is equivalent (up to machine precision) to the proposed method (Fig. 3 loss function decay), while significantly accelerating the training process (Fig. 4).

**Weaknesses:**

While the paper presents an original contribution, several issues need to be addressed to significantly enhance its impact, particularly in the experimental section, to convincingly demonstrate that the proposed approach is worth exploring. Below I summarize the key points:


**Poor literature review**: The literature review in the paper focuses primarily on loss optimization strategies and does not adequately cover methodologies addressing the key issues the paper aims to solve, namely memory efficiency, sequential execution bottlenecks, and redundant graph traversals. I suggest the authors enhance this section by including more relevant works. For instance, [1,2] explore numerical schemes for computing differential operators, while [3] employs forward-mode automatic differentiation to scale PINN training for a high number of collocation points and input dimensions. Additionally, the development of open-source software specifically designed for highly optimized PINN training I think should be mentioned, for example [4,5].

**Limited experiment section**: The experimental evaluation is restricted to only two simple PDEs, the heat and wave equations, which can be solved easily using PINNs in a short amount of time (as also noted by the authors). To strengthen claims regarding computational improvements, I suggest the following:
1. Train PINNs on more challenging PDEs, e.g., the (2+1)-D Navier-Stokes equation (see Appendix D.4 of [3]) or the Gray-Scott equation (see Section 5.3 of [6]), to provide stronger evidence of efficiency.
2. Report the average GPU memory usage for both the proposed method and the baseline PINN, and evaluate how memory usage scales with an increasing number of collocation points.
3. Assess the robustness of your method by repeating experiments at least four times with different random seeds and reporting the variance in the metrics.

**Applications to multiple activation functions**: To demonstrate the generality of your method, more activation functions should be tested, e.g., tanh, sigmoid, or SiLU. This would strengthen the paper by showing broader applicability. For example, the ELU activation function is limited when solving PDEs/ODEs with derivatives higher than second order. Specifically, for $x \ge 0$, all higher-order derivatives vanish: $\text{ELU}^{(n)}(x) = 0, \quad n \ge 2$, while for $x < 0$, all higher-order derivatives are identical: $\text{ELU}^{(n)}(x) = \alpha e^x, \quad n \ge 2$. This means that the network higher order derivatives will be the same as second order derivatives, which could result in networks not solving ODE/PDE with higher order (>2) terms in the differential operator.

**Minor issues**:
1. Line 168 should read "with $\mathbf{h1}$" (with a space), and similarly for lines 227, 247, 248, 266, and 313.
2. Line 422 is missing a period after "Figure 4."
3. The Adam optimizer is mentioned but not cited.


***References***

[1] Sharma, Ramansh, and Varun Shankar. "Accelerated training of physics-informed neural networks (pinns) using meshless discretizations." Advances in neural information processing systems 35 (2022): 1034-1046.

[2] Chiu, Pao-Hsiung, et al. "Marrying the benefits of Automatic and Numerical Differentiation in Physics-Informed Neural Network.", Workshop *Machine Learning and the Physical Sciences*, Advances in neural information processing systems 38 (2025).

[3] Cho, Junwoo, et al. "Separable physics-informed neural networks." Advances in Neural Information Processing Systems 36 (2023): 23761-23788.

[4] Lu, Lu, et al. "DeepXDE: A deep learning library for solving differential equations." SIAM review 63.1 (2021): 208-228.

[5] Coscia, Dario, et al. "Physics-informed neural networks for advanced modeling." Journal of Open Source Software 8.87 (2023): 5352.

[6] Wang, Sifan, et al. "Piratenets: Physics-informed deep learning with residual adaptive networks." Journal of Machine Learning Research 25.402 (2024): 1-51.

**Questions:**

1. Can you provide a mathematical explanation for why the memory usage is lower? Section 2.3 could benefit from a more in-depth analysis comparing your method with standard automatic differentiation (AD).
2. Is the approach easily implementable on modern architectures without requiring a complete rewrite each time? Including a modular pseudo-code algorithm would help illustrate how to integrate the method in practice.
3. How does your method scale with an increasing number of collocation points and higher-order derivatives?

---

### Official Review · Reviewer_6Sgs · 2025-10-30

**Soundness:** 2
**Presentation:** 1
**Contribution:** 3
**Rating:** 2
**Confidence:** 3

**Summary:**

This paper introduces the idea of computing all needed derivatives of a neural network in a single forward pass, by relying on specific properties of the ELU activation function. The paper applies this to PINNs. Experiments are done on two PDEs, 2D heat and wave equations. In these settings, a 1.5-2x speedup is observed by avoiding the need for backward passes to compute multiple derivatives.

**Strengths:**

* The central idea of the paper is simple and clear -- modify the activation function of the neural network to have a property such that derivatives can be computed in forward passes. ELU is an example that satisfies this.
* Numerical verification of this approach is done.
* Experiments on PINN problems show speedup over standard backprop for computing derivatives.
* Properties of this algorithm are well suited to modern GPUs (more compute intensity).

**Weaknesses:**

* The idea is simple and effective, but the gap to related work (forward mode AD) does not seem that high. It seems like a specific architectural choice that makes forward mode AD more efficient.
* The presentation is lacking and clarity could be greatly improved. For example:
   - The main result hinges on the ELU activation, but nowhere is the activation function defined. While the appropriate paper is cited, it would drastically improve the flow and exposition of the paper to include the basic definition in the introduction of the main idea.
   - Some of the notation is unnecessarily unclear. For example, in line 89-90, it seems like act is basically the sign function? This adds unnecessary complications in exposition.
   - The exposition directly presents derivatives of ELU without defining the original ELU function. Further, it would be greatly illuminating to extract out exactly what property of ELU is needed to enable this forward-mode computation of derivatives.
* The scope of experimental evaluation is quite limited. Only two PDEs for a specific configuration is evaluated.
* The paper would also greatly benefit from a more rigorous analysis of the gains in performance from the higher arithmetic intensity of the new algorithm on GPUs.

**Questions:**

1. Can the authors elaborate on previous work on forward mode AD approaches, and how their central idea improves on existing literature?
2. Could the authors encapsulate the essential property of the activation function needed to enable their approach to work?
3. Could the authors comment on numerical stability issues (if any) in the proposed approach?

---

### Official Review · Reviewer_BuXq · 2025-10-31

**Soundness:** 3
**Presentation:** 2
**Contribution:** 2
**Rating:** 2
**Confidence:** 4

**Summary:**

The paper introduces Forward PINN, a method for Physics-Informed Neural Networks that computes both PDE solutions and all the required spatial derivatives in a single forward pass instead of requiring multiple backwards passes. The authors exploit the analytical properties of the ELU activation function to obtain mathematically exact derivatives with a recursive algorithm. On benchmark PDEs, Forward PINN attains 1.55$\times$ speedup on the 2D heat equation and 1.8$\times$ speedup on the anisotropic wave equation, with solution accuracy comparable to a conventional PINN.

**Strengths:**

- The approach of using forward AD for computing PDE losses in PINNs is novel, as far as I am aware. Furthermore, the problem is important, as avoiding multiple backwards passes directly translates to speedups during training.
- The paper is generally clearly written, including complete descriptions of the method and the experimental setups. The approach, exploiting the properties of ELU, is elegant and clever.

**Weaknesses:**

- The experimental results are generally limited.
  - The forward PINN is only evaluated on two problems (2D heat equation and anisotropic wave equation), both of which are relatively simple problems. To make the results more thorough, evaluating on stiff PDEs (Burgers, K-S) or Navier-Stokes [1] may be more convincing.
  - The only baseline is a conventional PINN whose architecture identically matches the architecture of the forward PINN. I find the results in Figure 3/Figure 4 very convincing that the Forward PINN does not lose any expressivity compared to the conventional PINN of the same architecture. However, the current experiments do not evaluate whether Forward PINN can match the performance of modern PINN architectures, e.g. PirateNet [2]. For example, it is unclear whether requiring ELU activation restricts the max achievable precision of Forward PINN.
  - To further strengthen the evidence for the wall-clock speedup, it would be useful to benchmark the Forward PINN on different GPU architectures, batch sizes, and problem sizes. Training for longer than 50 epochs may also be more convincing.

1. Wang et al. An Expert's Guide to Training Physics-informed Neural Networks.
2. Wang et al. PirateNets: Physics-informed Deep Learning with Residual Adaptive Networks

**Questions:**

- Is this method equivalent to analytically performing forward-mode AD on PINNs with ELU activation? If so, did the authors benchmark Forward PINN against torch's built-in forward-mode AD implementation with the same neural network architecture?
- Can the authors explain whether the same techniques may apply to PINN architectures without the ELU activation (e.g. tanh) or whose architectures differ from a fully-connected network (e.g. networks with Random Fourier Features [1] in the input layer).

1. Wang et al. An Expert's Guide to Training Physics-informed Neural Networks.

---

### Note · Program_Chairs · 2026-01-17
**Submission Desk Rejected by Program Chairs**

The following references in this submission do not refer to real documents and/or have major errors in bibliographic information:

 Christian H Bischof, Alan Carle, Peyvand Khademi, and Andrew Mauer. Automatic differentiation for inverse problems in x-ray astronomy. Computational Science–ICCS 2008, pp. 584–593, 2008.